# SNAP: TESTING THE EFFECTS OF CAPTURE CONDITIONS ON FUNDAMENTAL VISION TASKS

## ABSTRACT

Generalization of deep-learning-based (DL) computer vision algorithms to various image perturbations is hard to establish and remains an active area of research. The majority of past analyses focused on the images already captured, whereas effects of the image formation pipeline and environment are less studied. In this paper, we address this issue by analyzing the impact of capture conditions, such as camera parameters and lighting, on DL model performance on 3 vision tasks— image classification, object detection, and visual question answering (VQA). To this end, we assess capture bias in common vision datasets and create a new dataset, SNAP (for **S**hutter speed, ISO se**N**sitivity, and **AP**erture), consisting of images of objects taken under controlled lighting conditions and with densely sampled camera settings. We then evaluate a large number of DL vision models and show the effects of capture conditions on each selected vision task. Lastly, we conduct an experiment to establish a human baseline for the VQA task. Our results show that computer vision datasets are significantly biased, the models trained on this data do not reach human accuracy even on the well-exposed images, and are susceptible to both major exposure changes and minute variations of camera settings.

## 1 INTRODUCTION

Data is one of the pillars of modern deep-learning-based (DL) computer vision as training and evaluation of models rely on large image datasets. Generalization beyond training data is highly desirable but difficult to determine for current DL models Nagarajan & Kolter (2019); Dziugaite et al. (2020); Zhang et al. (2021); Chatterjee & Zielinski (2022). Several factors contribute to this, including DL model opacity combined with lack of established DL theory Goldblum et al. (2020); He & Tao (2020); Suh & Cheng (2024) and the growing volume of training data Qin et al. (2024). Past research on DL generalization focused on measuring and mitigating data biases and sensitivity of DL models to input perturbations, such as adversarial examples Wei et al. (2024), image distortions Hendrycks & Dietterich (2019), viewing angles Barbu et al. (2019), incongruous context Hendrycks et al. (2021), etc. So far, most analyses focused on images already captured but not the image formation process, which has many variables that depend on the sensor, its settings, and environment properties. If any of these elements are altered, images of the same scene may appear very differently. However, the effects of these changes on vision algorithms have not been systematically examined.

In this paper, we focus on the effects of capture conditions, i.e. camera parameters and lighting, on 3 fundamental visual tasks—image classification, object detection, and visual question answering (VQA). To do so, we 1) analyze capture bias in the existing vision datasets; 2) gather a new dataset, SNAP, where capture bias is minimized by dense sampling of camera parameters under controlled lighting conditions; 3) estimate the effects of capture settings on vision tasks by testing a representative set of models on SNAP; 4) associate model performance with the properties of the training data and models themselves; and 5) conduct a human study to establish a human baseline for common vision tasks under different capture conditions.

## 2 RELATED WORKS

**Dataset biases.** Most vision datasets are biased, which affects the representations and generalization ability of the trained algorithms. Classification datasets, in particular, are found to be biased in terms

of object classes, sizes, backgrounds, and viewpoints Ponce et al. (2006); Torralba & Efros (2011); Herranz et al. (2016); Tommasi et al. (2017); Azulay & Weiss (2018); Tsotsos & Luo (2021); Zeng et al. (2024); Liu & He (2025). To our knowledge, only a few works considered sensor parameter bias (referred to as *capture bias* in Tommasi et al. (2017)) in computer vision data. These include studies of consumer photos Wueller & Fageth (2008; 2018), analyses of VOC2007 Everingham et al. (2010) and COCO Lin et al. (2014) metadata in Tsotsos et al. (2019), and common Exif tags for YFCC100M Thomee et al. (2016) reported in Zheng et al. (2023).

**Effect of image degradations on algorithms.** Numerous studies investigated the effects of artificial image degradations on the performance of classical and DL vision and vision-language algorithms. To facilitate this, widely used datasets have been modified with artificial corruptions and perturbations, starting with ImageNet-C Hendrycks & Dietterich (2019), followed by MNIST-C Mu & Gilmer (2019), as well as Pascal-C, COCO-C, and Cityscapes-C Michaelis et al. (2019). Some of the common distortions considered in prior works are blur Dodge & Karam (2016; 2017); Grm et al. (2017); Roy et al. (2018); Che et al. (2019); Hendrycks & Dietterich (2019); Chen et al. (2023a); Yamada & Otani (2022), various types of noise Le Meur (2011); Dodge & Karam (2016); Liu et al. (2016a); Niu et al. (2016); Rodner et al. (2016); Tow et al. (2016); Rodner et al. (2016); Dodge & Karam (2017); Grm et al. (2017); Roy et al. (2018); Geirhos et al. (2018); Hendrycks & Dietterich (2019); Chen et al. (2023a); Yamada & Otani (2022), digital artifacts Le Meur (2011); Dodge & Karam (2016); Zheng et al. (2016); Roy et al. (2018); Che et al. (2019); Hendrycks & Dietterich (2019); Chen et al. (2023a); Yamada & Otani (2022), changes in brightness Tow et al. (2016); Grm et al. (2017); Hendrycks & Dietterich (2019), contrast Dodge & Karam (2016); Grm et al. (2017); Geirhos et al. (2018); Che et al. (2019); Hendrycks & Dietterich (2019), color Rodner et al. (2016); Geirhos et al. (2018), etc.

While blur, noise, and other distortions may also be caused by changing camera settings Tow et al. (2016); Roy et al. (2018); Bielova et al. (2019); Che et al. (2019), only a few studies considered the effects of natural capture conditions. One of the early tests Andreopoulos & Tsotsos (2011) demonstrated performance fluctuations of classical vision algorithms on 4 scenes acquired with different shutter and gain settings under 3 illumination conditions. In Wu & Tsotsos (2017), classical and DL object detectors were tested on the set of 5 objects taken with 64 camera configurations and 7 lighting levels. Another study evaluated DL object detectors on subsets of COCO corresponding to different camera settings and showed performance variations on bins with different number of training samples Tsotsos et al. (2019). Most recently, the authors of Baek et al. (2024) collected a new dataset, ImageNet-ES by displaying 2000 images from the original ImageNet on the TV screen and photographing each across 64 camera parameter sets with and without external lighting. The dataset was used to show sensitivity of models to these changes and benefits of diversifying training data w.r.t. capture conditions.

**Effect of image degradations on human vision.** Several studies compared human and machine performance on various types of image degradations and tasks (e.g., fine-grained image classification Dodge & Karam (2017), saliency Che et al. (2019), and object recognition Geirhos et al. (2018); Shen et al. (2025)). Performance of both humans and CNNs was shown to be affected but to a different extent. Human performance declined significantly only for the highest distortion levels Geirhos et al. (2018); Shen et al. (2025). And while some deep networks outperformed humans on undistorted images (e.g., on fine-grained dog breed classification Dodge & Karam (2017)) they degraded faster under distortions Geirhos et al. (2018); Shen et al. (2025). Both humans and CNNs were relatively robust to minor color-related distortions Geirhos et al. (2018).

This paper expands previous work both in depth and scope: 1) we analyze Exif data of 13 common vision datasets, particularly focusing on those used for training and evaluation of models for object detection and classification and foundation vision models, 2) collect a dataset of 100 real scenes with over 700 densely sampled camera parameter combinations under 2 controlled illumination conditions, 3) test 52 models on classification, detection, and VQA tasks and collect data from 43 human subjects for the latter task, and 4) systematically analyze performance of algorithms w.r.t. capture conditions, compare it to humans, and link it to biases in the training data.

## 3 METHODOLOGY

This session discusses the paradigm, procedure, metrics, and models for probing DL vision models across capture conditions. Section 3.1 shows significant capture bias in common vision datasets

that motivates the balanced design of the SNAP datasetWe will add the clarification that even if CV does not reach > 1, there is still noticeable fluctuation of accuracy even in the well-exposed range for all models described in Section 3.2. Section 3.3 lists the models and evaluation metrics. Lastly, Section 3.4 is dedicated to the experiment conducted on a subset of SNAP to establish a human baseline for vision tasks under varying capture conditions.

### 3.1 ANALYSIS OF CAPTURE BIAS IN COMMON VISION DATASETS

We analyzed capture bias in over 1B images from the following popular datasets for training image classifiers, object detectors, and foundation models: VOC2007 Everingham et al. (2010), ImageNet Deng et al. (2009), SBU Ordonez et al. (2011), COCO Lin et al. (2014), YFCC15M Thomee et al. (2016), CC3M Sharma et al. (2018), ImageNet21K Ridnik et al. (2021), CC12M Changpinyo et al. (2021), WIT Srinivasan et al. (2021), LAION400M Schuhmann et al. (2021), OpenImages v7 Benenson & Ferrari (2022), COYO Byeon et al. (2022), and Wukong Gu et al. (2022).

Table 1 shows a summary of the datasets and metadata availability. Overall, metadata in the datasets is distributed highly unevenly; datasets originating from the Common Crawl generally contain much fewer images with metadata (1-6%) than datasets from curated resources, such as Flickr and Wikipedia (30-60%).

We extracted Exif information from images belonging to these datasets. We used the Flickr API to gather metadata for datasets with images from Flickr. For the rest, we downloaded the images and extracted Exif tags using the ExifTool Harvey (2016). Because many recent datasets distribute only image URLs, not all images could be downloaded and analyzed due to broken links.

We focus on the basic camera settings (aperture, shutter speed, and ISO) and program mode (auto or manual) because other tags, such as lighting, focal distance, white balance, etc., are absent for most images. Overall, we found significant biases across all datasets. The distribution of camera settings is very long-tailed with distinct peaks at several F-number (2.8, 4, 5.6, 8), ISO (100, 200, 400), and shutter speed (1/125, 1/60, 1/30) values (see Fig. A.4). Since 50-80% of all images were taken with auto settings (Fig. A.1), we can infer that most

Table 1: Availability of images and metadata in the common vision datasets. "w/ Exif"—all 3 tags of interest (shutter speed, F-number, and ISO) are in metadata.

| Dataset | Year | # images | # downloaded | # w/ Exif |
|---|---|---|---|---|
| VOC2007 Everingham et al. (2010) | 2007 | 10.0K | 10.0K (100%) | 3.0K (29.7%) |
| ImageNet Deng et al. (2009) | 2009 | 1.6M | 1.6M (100%) | 59.7K (3.8%) |
| SBU Ordonez et al. (2011) | 2011 | 1.0M | 749.3K (74.9%) | 492.3K (49.2%) |
| COCO Lin et al. (2014) | 2014 | 164.0K | 164.0K (100%) | 68.6K (41.8%) |
| OpenImages v7 Benenson & Ferrari (2022) | 2016 | 1.9M | 1.9M (100%) | 1.1K (0.1%) |
| YFCC15M Thomee et al. (2016) | 2016 | 15.4M | 13.0M (84.6%) | 9.5M (61.6%) |
| CC3M Sharma et al. (2018) | 2018 | 3.3M | 2.3M (70.0%) | 105.1K (3.2%) |
| ImageNet21K Ridnik et al. (2021) | 2021 | 13.2M | 13.2M (100%) | 797.0K (6.1%) |
| CC12M Changpinyo et al. (2021) | 2021 | 12.4M | 8.1M (64.9%) | 280.5K (2.3%) |
| WIT Srinivasan et al. (2021) | 2021 | 26.5M | 25.9M (97.8%) | 12.1M (45.7%) |
| LAION400M Schuhmann et al. (2021) | 2021 | 413.6M | 304.4M (73.6%) | 5.8M (1.4%) |
| Wukong Gu et al. (2022) | 2022 | 101.4M | 96.0M (95.0%) | 125.2K (0.1%) |
| COYO Byeon et al. (2022) | 2022 | 747.0M | 512.9M (68.7%) | 15.3M (2.1%) |
| Total | - | 1.3B | 980.1M (73.3%) | 44.7M (3.3%) |

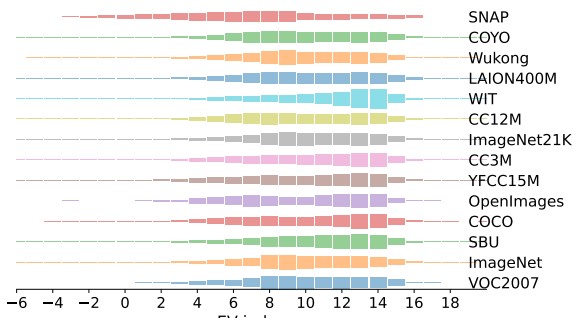

Figure 1: Normalized distribution of camera settings grouped by EV index for each dataset. Plots are arranged chronologically from bottom to top. Our SNAP dataset introduced in Section 3.2 is shown for reference.

of them are well-exposed. We can also group images with similar exposure by computing exposure values (EV) from camera parameters, as will be explained in Section 3.2. Distribution of the EVs across datasets shown in Fig. 1 indicates that most images were taken in well-lit indoor and outdoor spaces. Finally, there is a significant chronological bias in vision data (Fig. A.2); for instance, the bulk of images in ImageNet and COCO are from a decade ago and may no longer be representative of modern sensors and objects.

### 3.2 SNAP DATASET WITH BALANCED CAPTURE CONDITIONS

To systematically investigate the effects of capture conditions, we collected a novel dataset, SNAP (for **S**hutter speed, ISO se**N**sitivity, and **AP**erture). It contains photos of multiple scenes taken under controlled lighting conditions and balanced across camera parameters. We designed SNAP for evaluating the models on 3 vision tasks: image classification, object detection, and visual question answering (VQA). This section describes SNAP's design, collection procedure, and properties.

### 3.2.1 DATA COLLECTION AND PROPERTIES

**Capture setup.** Fig. 2 shows our data collection setup: a camera tethered to a laptop and placed on a tripod in front of a table covered with gray non-reflective cloth. Both the camera and the table are inside a blacked-out room where the only sources of light are 2 LED panels on either side of the table. To ensure consistent illumination, we use Yocto-Light-V3 lux meter Yoctopuce. All images are taken with Canon EOS Rebel T7 DSLR camera and saved as $1920 \times 1280$ JPEG files.

On the camera side, we modify the shutter speed, F-Number, and ISO at 1-stop intervals within the available range of settings (see Table 2). We use two illumination conditions, 1000 and 10 lux, roughly corresponding to an overcast day and twilight, respectively. Use of both high and low illumination allows expanding the range of usable camera settings. For example, under the low lighting conditions images can be taken with longer exposure times and higher ISO values. Other camera settings are fixed: flash is not used, focal length is set at 35 mm, and white balance is configured manually and fixed for each illumination condition.

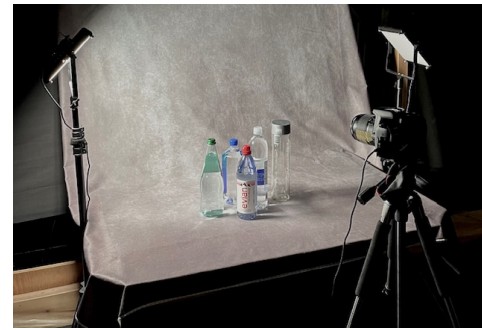

Figure 2: Data collection setup consists of a Canon EOS Rebel T7 camera on the tripod, a table covered with non-reflective cloth backdrop, and LED lights on either side.

**Object categories.** Dense sampling of camera parameters is time-consuming to collect. Thus to make the data suitable for all three vision tasks, we chose object categories present in both ImageNet and COCO—the most common datasets for training image classification and object detection/VQA models, respectively. SNAP contains the following 10 object categories that provide sufficient visual variety and physically fit within our setup: backpack, tie, water bottle, cup, laptop, mouse, remote, keyboard, cell phone, and comic book. Using only inanimate objects avoids motion artifacts during long exposure times and maintains layout constant throughout the capture. As our goal was to investigate the effects of camera settings, we minimized other effects: objects were photographed in canonical object poses (as models are sensitive to poses Barbu et al. (2019)) and against a neutral background (to prevent shortcuts Geirhos et al. (2020)).

In all images, there are 2 to 5 objects of the same class but with different appearance (e.g. a scene with 5 different water bottles in Fig. 2). Having multiple objects in the scene adds minor occlusions, shadows, and clutter that in turn makes object detection and VQA tasks more realistic. Since all objects are of the same category, multi-label errors (common in the ImageNet Stock & Cisse (2018); Tsipras et al. (2020); Beyer et al. (2020)) are avoided.

Table 2: Canon EOS Rebel T7 camera settings used for data collection.

| Camera parameters | Values |
|---|---|
| Shutter speed (s) | 1/4000, 1/2000, 1/1000, 1/500, 1/250 1/125, 1/60, 1/30, 1/15, 1/8, 1/4, 0.5, 1, 2, 4, 8, 15, 30 |
| ISO | 100, 200, 400, 800, 1600, 3200, 6400 |
| F-Number | 5.6, 8, 11, 16, 22 |

**Capture procedure.** For each scene and lighting condition, we first take one image with a camera's auto setting, then switch the camera to manual mode and use the gPhoto2 library to capture images over all possible combinations of F-number, shutter speed, and ISO sampled at 1-stop intervals. Photos with more than 95% black (0) or white (255) pixels are discarded. One iteration through camera parameters for one lighting condition takes 40–60 minutes.

**Data composition and properties.** We capture 10 scenes with 2–5 objects from each object category (shown in Fig. B.1. There are a total of 37,558 images in the dataset, uniformly distributed across sensor settings. However, many combinations of shutter speed, aperture, and ISO lead to the same exposure, i.e., the same amount of light reaching the sensor (see examples in Fig. B.2). This is known as exposure equivalence (Präkel, 2009, p.29). To identify such settings, we compute exposure value (EV) for each triplet using a standard formula Ray et al. (2000). Because exposure depends also on the lighting conditions, we group together settings with the same EV and lux values. We then re-index EV bins so that best exposure settings (i.e. closest to camera auto mode) receive a value of 0, over-exposed bins receive positive indices, and under-exposed are assigned negative ones. We refer to these indices as EV offset. For example, an EV offset of -1 means that the image is one stop underexposed, i.e., received half the light needed for the best exposure (see Fig. B.3).

### 3.2.2 ANNOTATIONS AND VLM PROMPTS

All images in SNAP are provided with Exif information, object class label, bounding boxes, and segmentation masks. To test VLMs on image classification and object detection, we generated question-answer pairs from the annotations. Because selected VLMs lack object detection abilities, we use counting (subitization) as a proxy since it requires localization. Due to the prompt sensitivity of VLMs Li et al. (2024), each question has open-ended (OE) and multi-choice (MC) variants:

**Open-ended**
**Q1.** Objects of what class are in the image? Answer with the name of the class.
**Q2.** How many objects are in the image? Answer with one number.
**Multiple choice** (answer options shuffled for each image)
**Q3.** Objects of what class are in the image? Select one of the following options: 10 classes + other
**Q4.** How many objects are in the image? Select one of the following options: A) 2; B) 3; C) 4; D) 5

### 3.3 MODELS AND METRICS FOR FUNDAMENTAL VISION TASKS

We tested baseline and SOTA models, including 23 image classifiers, 16 object detectors, and 13 vision-language models (VLMs), on image classification, object detection, and visual question answering (VQA) tasks, respectively (see Appendix D). We selected only open-source VLMs to be able to analyze their performance w.r.t. their training data and vision backbones. All selected models were tested on SNAP with default parameters, as well as pre- and post-processing routines where applicable. All experiments were performed on a cluster of 4 NVIDIA GeForce GTX 1080 Ti. The following metrics are used for evaluation:

- **Top-1 (%)**—a standard top-1 accuracy metric for image classification;

- **Localization-Recall-Precision (LRP)** metric Oksuz et al. (2021; 2018) for object detection. Unlike commonly used average precision (AP), LRP decouples classification and localization performance of the models. We report on the optimal LRP (oLRP), as well as localization accuracy (oLRP Loc), false positives (oLRP FP), and false negatives (oLRP FN).

- **Soft and hard accuracy** for the VQA task. Soft accuracy is % of answers that partially match the ground truth. Hard accuracy rewards answers that are both factually correct and faithful. Following Huang et al. (2025), the answer is considered a factual error if it refers to something not present in the image and a faithfulness error if the answer does not adhere to the required format. See Appendix E for details.

- **Parameter sensitivity (PS)** measures sensitivity of the metrics to camera parameters on all tasks. As discussed in Section 3.2, SNAP contains many images of the same scenes that look nearly the same but are taken with different camera settings under different lighting levels (Fig. B.2). To evaluate whether these barely perceptible changes affect model performance, we do the following: 1) we group the images of the same scene with the same EV offset; 2) to measure fluctuations in models' results within each set, we compute coefficient of variation (CV), defined as the ratio of the std to mean ($CV = \frac{\sigma}{\mu}$) of the metric; 3) we compute PS as the percentage of sets with $CV > 1$.

### 3.4 HUMAN EXPERIMENT

To establish a human baseline under varying capture conditions, we use the VQA task described in Section 3.2.2 because it includes image classification (Q5), counting as a proxy for object detection (Q3), and allows direct comparisons to VLMs. We tested 43 human subjects on a subset of 8600 images from SNAP, evenly distributed w.r.t. object categories and capture conditions. The experiment was run in-person in the blacked-out room because online platforms (e.g. Amazon Turk) do not provide means to control lighting, display calibration, and presentation times, which are crucial for our study. During the experiment, the subjects viewed the images sequentially on the computer screen and for each answered either the categorization or counting questions listed in Section 3.2.2. Since we intended to compare human answers to outputs of the predominantly feedforward DL vision models, the study was designed such that feedback processing in the brain was minimized. To achieve this, stimuli were shown very briefly (200 ms) and were immediately followed by the noise mask (200 ms), as in the past studies Geirhos et al. (2018). See Appendix C for full details.

# 4 EXPERIMENT RESULTS

## 4.1 IMAGE CLASSIFICATION

**Most models do not generalize to SNAP despite its simplicity and similarity to ImageNet.** As shown in Fig. 3, only 2 models (CLIP ViT-L/14@336px, DFN CLIP ViT-L/14) reach mean top-1 accuracy on SNAP comparable to their performance on ImageNet, while others show a significant decrease in accuracy ranging from 10 to 50%.

Fig. F.1b and Fig. F.1a show plots of top-1 accuracy on SNAP relative to the model and training data size, respectively. Scaling both models and training data improves the results; all models with performance above 80% top-1 accuracy on SNAP have at least 200M parameters and are trained on 400M images or more.

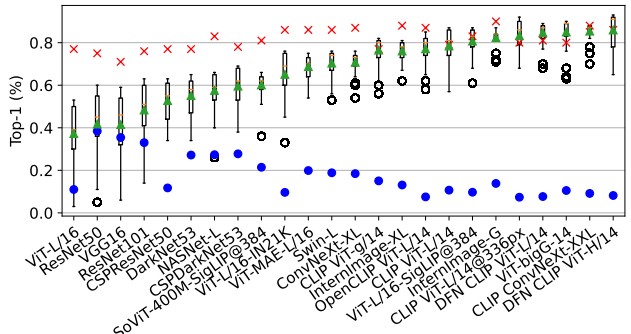

The effects of the architecture and training are less pronounced. While 4 out of top-5 models are Transformers, some CNN-based models achieve competitive results (see Fig. 3). For example, supervised training of vanilla ViT L/16 and VGG-16 on ImageNet results in similar performance and so does CLIP-pretraining of ViT and ConvNeXt on comparable amounts of data.

Models trained or fine-tuned on ImageNet struggle with some categories in SNAP. For instance, most models misclassify cups and laptops as mugs and notebooks, respectively, due to inconsistent labeling in ImageNet. Hence, in evaluation, we accept both answers as correct. Most models also underperform on phones, which is the consequence of the chronological bias pointed out earlier in Section 3.1. Because most images in ImageNet were taken prior to 2009, it is dominated by photos of outdated phone designs. Only the zero-shot CLIP-pretrained models (not finetuned on ImageNet) were able to classify modern phones in SNAP correctly.

**Models reach peak top-1 accuracy on well-exposed images but perform**

Figure 3: Box plots show range and mean top-1 accuracy values for all models evaluated on SNAP. Blue circles show parameter sensitivity (PS) and red crosses mark top-1 accuracy on ImageNet. Black circles represent outliers.

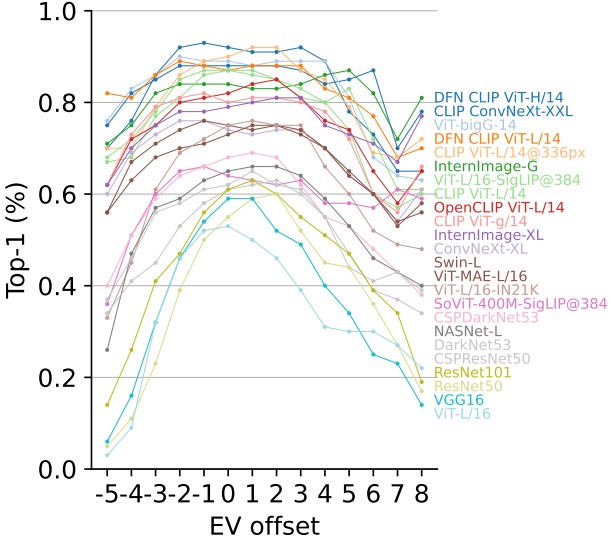

Figure 4: Image classification top-1 (%) across exposure levels in SNAP. Each dot represents mean accuracy across all images in the corresponding EV offset bin.

**significantly worse on under- and over-exposed images.** Exposure, i.e. the combined effect of camera settings and illumination, affect all models (see Fig. 4). The models reach their peak performance on well-exposed images (EV offset between -2 and 2). 8 out of top-10 models match or surpass their ImageNet results on this subset of SNAP, which is expected given the intentional similarity of the objects and scenes. At the same time, all models perform worse on over- and under-exposed images. This performance drop is not symmetric; accuracy on the under-exposed images is lower than on the over-exposed images.

Again, the larger models (regardless of the architecture) trained on more data achieve higher peak accuracy on well-exposed images and are less affected even by the most extreme exposures. For

instance, CLIP-pretrained ConvNeXt-XXL reaches over 70% accuracy on nearly all-black (EV offset -5) and all-white (EV offset 8) images. Notably, this model is trained on the 2B subset of LAION with minimal data augmentations (only random cropping and erasing). This agrees with our analysis in Appendix A that shows the higher diversity of manual settings and exposures in LAION.

**All models are susceptible to image perturbations caused by slight variations of camera parameters.** This is evident from high parameter sensitivity (PS) w.r.t. top-1 accuracy that reaches over 20% some models (Fig. 4). Even the top-5 performing models inconsistently classify nearly 10% of the scenes that look essentially the same. Even when CV does not reach > 1, the accuracy of all models fluctuates even in the well-exposed range.

## 4.2 OBJECT DETECTION

**Many models perform well overall, but SNAP is simpler than most object detection datasets.** Fig. 5 shows the results of testing object detection models on the SNAP dataset. For this experiment, we also computed the mean AP scores of the models to compare against the COCO benchmark results. The mean AP of the models is approx. 15-20% higher on SNAP (Fig. F.2). This can be explained by the relative simplicity of SNAP: the scenes are not cluttered, objects appear large against plain background, and there is little occlusion or cropping (see Fig. B.1). In comparison, the scenes in COCO are busy, filled with multiple small objects, many of which are occluded or cropped.

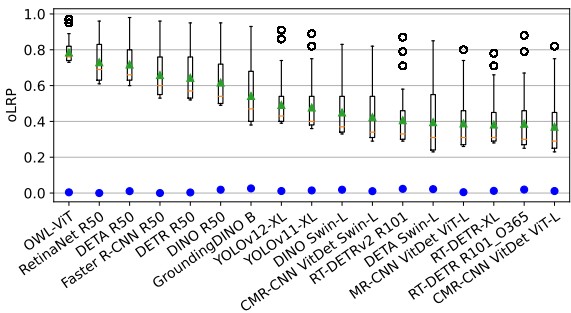

Figure 5: Box plots show range and mean oLRP values for all object detection models evaluated on SNAP. Blue circles show parameter sensitivity (PS) w.r.t. oLRP. Black circles represent outliers.

The majority of existing object detection models are pre-trained on ImageNet and COCO, therefore the effects of scaling data cannot be explored fully. There is some evidence of positive correlation between model scale and performance. For instance, the top model DETA Swin-L is trained on the Object365 dataset Shao et al. (2019) with 600K images and 10M bounding boxes in addition to COCO. At the same time, RT-DETR is trained only on COCO and performs on par.

**Effects of exposure on object detection are more pronounced than on image classification.** Similar to image classifiers, the overall performance (oLRP) of object detectors degrades on under- and over-exposed images, as illustrated in Fig. 6. In this case, however, the drop-off towards either extreme is more pronounced and is apparent even in the well-exposed range for all models. There is also asymmetry in the oLRP scores, but the trend is reversed—unlike image classifiers, the object detection models perform worse on the over-exposed images.

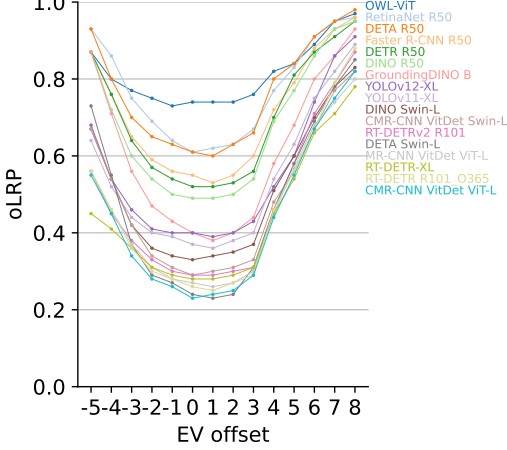

Figure 6: Object detection at different exposure levels in SNAP. Each point is a mean oLRP of the model predictions on a given EV offset. Model labels are sorted by oLRP from worst to best)

Exposure affects both localization and classification, but to a different extent. Localization (oLRP Loc) and false positive (oLRP FP) errors increase for under- and over-exposed images, but remain relatively low for all models (see Fig. F.3a and Fig. F.3b). Misclassifications (oLRP FN) are the largest contributor to the overall oLRP score. They mirror the pattern of top-1 accuracy on the image classification task: errors are fairly low for well-exposed images but rise quickly toward larger EV-offsets in both directions (as shown in Fig. F.3c). The majority of object detectors use ImageNet

pre-trained backbones (typically, variants of ResNet or ViT) and are further trained only on COCO for the object detection task. Therefore, misclassification errors are likely an artifact of ImageNet pre-training. Several exceptions from this trend lend further support to this conclusion. Three models (DETA Swin-L, DINO Swin-L, and Grounding DINO) with consistently low oLRP Loc and oLRP FP scores across all EV offsets are all trained on ImageNet21K and O365. Similarly, OWL-ViT, which uses a large CLIP ViT backbone and is trained on O365 Shao et al. (2019) and VG Krishna et al. (2017) for detection, has a flatter curve (albeit at a higher error), reminiscent of CLIP ViTs in Fig. 4.

**Most of the fluctuations on images with similar exposure come from misclassifications (FN errors), whereas localization is less affected.** While the overall sensitivity of the oLRP scores for all models is low, localization and classification performance is still affected. For nearly all models, SP is highest for the oLRP FN component, reaching nearly 20% for some.

### 4.3    VISUAL QUESTION ANSWERING

**VLMs are comparable to humans in average accuracy across all questions.** We first assess the average performance of the tested VLMs against the human subjects by aggregating accuracy across all images and questions. Computing the accuracy scores for VLMs is a challenge. In many cases, matching the answer of the models with the ground truth may inflate the scores because not all models follow the prompt well. We used a combination of simple regular expressions and manual clean-up to bring all model answers to the same format (see Appendix E).

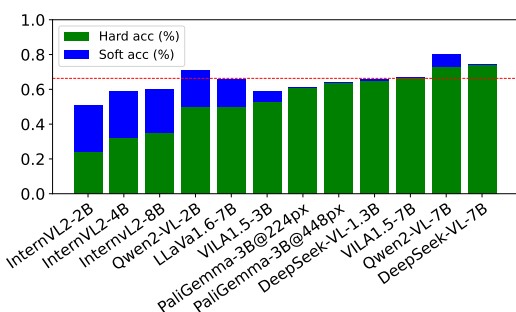

Figure 7: Mean hard accuracy (green) and soft accuracy (blue) on all questions for models and human subject accuracy (red line).

Soft and hard accuracy scores in Fig. 7 show percentage of partially correct and correct answers. The majority of the partially correct answers are due to not following the prompt. Although the questions explicitly ask to answer with a single number or multi-choice option (see Section 3.2.2), many VLMs instead return multiple categories of objects or generate a long explanation.

We used model answer length exceeding average expected answer (10-15 characters for most questions) as a proxy for these types of mistakes and ability of the models to follow the prompt (see Fig. 8). For example, PaliGemma, VILA, and DeepSeekVL-7B almost never deviate from the answer format, as do human subjects, whereas all InternVL models and Qwen2-VL-2B produce lengthy responses, unprompted chain of thought explanations, hallucinate non-existing answer options, and even switch between languages. Overall, there is a trend towards fewer hallucinations in models with larger LLM component. For instance, Qwen2-VL-7B

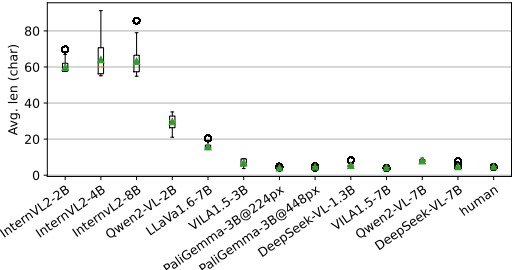

Figure 8: Mean length of responses (number of characters) across all questions. Longer answers indicate presence of faithfulness errors.

hallucinates substantially less compared to its 2B-parameter version. Similarly, InternVL2-8B is not as prone to hallucinations as 4B and 2B models from the same family.

Discounting partially correct answers, 4 models reach or surpass the average accuracy of human subjects (66%): Qwen2-VL-7B-Instruct (77%), DeepSeek-VL-7B-chat (74%), VILA1.5-7b (68%), and DeepSeek-VL-1.3B-chat (66%). All three use the largest vision models trained on the most data. DeepSeek and VILA use ViT-L-16-SigLIP-384 trained on 100B image pairs from the proprietary Webli dataset Chen et al. (2023b), whereas Qwen2-VL uses DFN5B-CLIP-ViT-H-14 vision encoder, which ranked first on the image classification task in Section 4.1. An interesting exception is PaliGemma: on one hand, its hallucination rate is near zero, on the other, its vision backbone, a shape optimized SigLIP ViT (SoViT-400M-SigLIP) pre-trained on Webli, is weaker than others. On SNAP, it achieves only 60% top-1 accuracy vs. 81% of the SigLIP-ViT variant used by DeepSeek and VILA.

**VLMs are less affected by under-exposed images than humans but do not reach human peak performance on well-exposed images.** We also looked at the performance of the models at different exposure levels. As in other tasks, performance of the models reaches its peak on the well-exposed images and drops off outside of that range. Despite lower overall results, human subjects reached a peak mean accuracy of 89% across all questions on well-exposed images (EV offset of 1), whereas the top-3 VLMs mentioned above were at 80%, 78% and 77%, respectively. Only on the MC categorization question, 4 top VLMs outperformed peak human accuracy (98.3%) on well-exposed images.

The low average performance of the human subjects is mainly due to sharp drop-off on under-exposed images (EV offset < -3). These images appear mostly black so the performance of the human subjects reduces to chance. On open-ended questions, many subjects explicitly indicated not being able to answer. While some models do that too (e.g. PaliGemma responded "unanswerable" in such cases), they are not consistent. On the over-exposed images, however, the human subjects maintained high level of performance surpassing nearly all VLMs. Exposure affects the rate of VLM hallucinations as well, whereas the human responses remain the same throughout. Most models tend to produce longer answers outside the well-exposed range, except Qwen models that do the opposite.

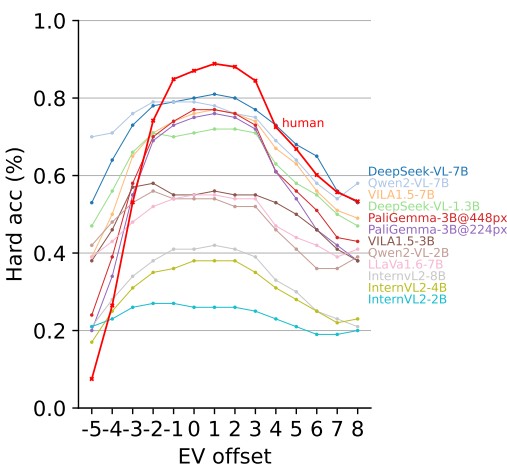

Figure 9: Mean hard accuracy on all questions and exposures in SNAP for VLMs and humans.

**All VLMs are sensitive to camera parameter variations on all questions but performance changes are unpredictable.** Likely due to the interactions between the vision and language components, each model had different performance trends that varied significantly depending on the question type (categorization and counting) and options (multi-choice or open-ended), as shown in Fig. F.5. For example, the strongest overall model DeepSeek-VL-7B is very consistent on MC categorization (Q5), but reaches 10-20% PS on other questions. The second best model Qwen2-VL-7B has the lowest overall PS on OE categorization (Q3) compared to other models, while its PS on other questions is relatively high (15-20%).

## 5    CONCLUSIONS

We conducted a systematic study to determine the sensitivity of DL vision algorithms to capture bias, i.e. changes in both camera parameters and illumination. We first identified a significant bias in the training data—even datasets with millions of images are highly imbalanced in terms of camera settings. We then constructed a novel dataset SNAP with more uniform capture conditions. This dataset was used to test a number of models and human subjects on common vision tasks. Three main conclusions can be made from this study. First, despite the intentional simplicity of SNAP, most models did not match their performance on other benchmarks and few surpassed humans. This gap persisted even for the largest models trained on billion-scale data. Second, both models and humans were sensitive to deviations of exposure, but in different ways. This discrepancy and the fact that most models did not reach human performance even under optimal conditions again points to the generalization issues. In addition, pre-trained image classification backbones propagate their capture biases on downstream tasks. Third, all models were sensitive minute variations in camera settings on all tasks. This is especially concerning for practical applications, where precise control of camera or illumination conditions is not possible.

**Limitations.** The time-consuming process of gathering real-world data did not allow testing other aspects of capture bias, such as effects of sensor type. We used Canon DSLR for capture since it has a wide range of parameters and is a very common camera make in the vision datasets. Mobile phones and webcams are less represented and are more difficult to control but may be more relevant for practical applications. Furthermore, sensor settings were sampled at 1-stop intervals, rather than 1/2 or 1/3 stops, and with only 2 lighting conditions because otherwise capture would take prohibitively long (multiple hours per scene).

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

## A  COMPUTER VISION DATASET PROPERTIES

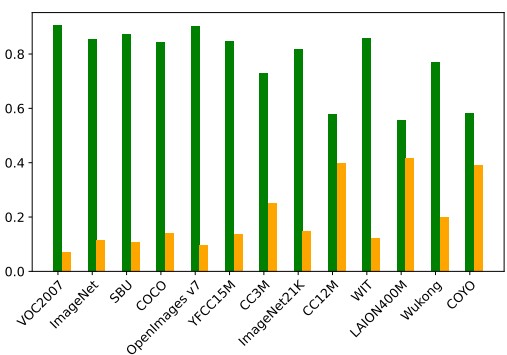

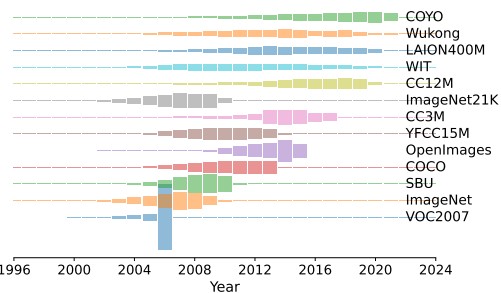

Figure A.1: Distribution of images in the datasets taken with auto (green bars) or manual (orange bars) camera settings.

Figure A.2: Distribution of images in the datasets by year the images were taken. Each bar plot corresponds to a single dataset. Plots are arranged in chronological order from bottom to top.

In this section we provide additional details on the properties of the common computer vision dataset. For this analysis we selected 13 common datasets that are used to train models for image classification, object detection, and fundamental vision models. The largest amount of data for our analysis comes from the latter category, namely datasets that contain image-text pairs collected from the Internet. We excluded several datasets, such as Object365 Shao et al. (2019) and VisualGenome Krishna et al. (2017), because all image metadata was stripped and sources for the images (e.g., URL links or Flickr ids) were not provided.

Fig. A.1 shows the distribution of images taken with automatic and manual camera settings. To identify mode, we check *ExposureMode* tags in image Exif data. The auto mode category includes various settings and presets that either fully or partially automate exposure. The most common tag values include: "Auto", 'Auto exposure', 'Aperture-priority AE', 'Auto bracket', 'Creative (Slow speed)', 'Shutter speed priority AE', 'Landscape', 'Portrait','Action (High speed)','Normal program', etc. Tag values corresponding to the manual mode are: "Manual" and 'Manual exposure". Overall, auto exposure was used to take 72% of all photos (with Exif data) in all datasets we analyzed and 26% of data was captured in manual mode.

We also looked at the distribution of values of 3 camera parameter values in the datasets. Fig. A.4 shows an example from the ImageNet, which is characteristic of the other datasets as all of them are very long-tailed and have distinct peaks at 1-stop intervals in all 3 settings.

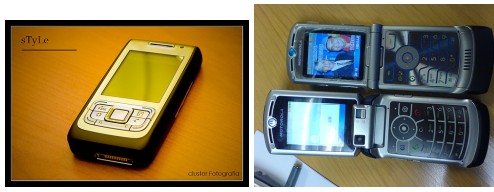

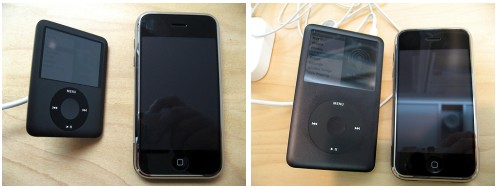

Figure A.3: Examples of chronological bias in ImageNet: images labeled as "phone" are on the left and "iPod" on the right.

Lastly, many models trained or fine-tuned on ImageNet had a much lower performance on the phone class (<30% top-1 accuracy). Even one of the top models CLIP ConvNeXt-XXL reached only 53% accuracy on this class. All models frequently misclassified phones as iPods. This issue likely stems from the chronological bias of ImageNet, where most images were taken more than a decade ago. Fig. A.3 shows examples from the *phone* and *iPod* synsets. Images for the phone synset are dominated by flip-phones and keyboard phones, whereas iPod class often features iPhones that look similar to the modern phone designs.

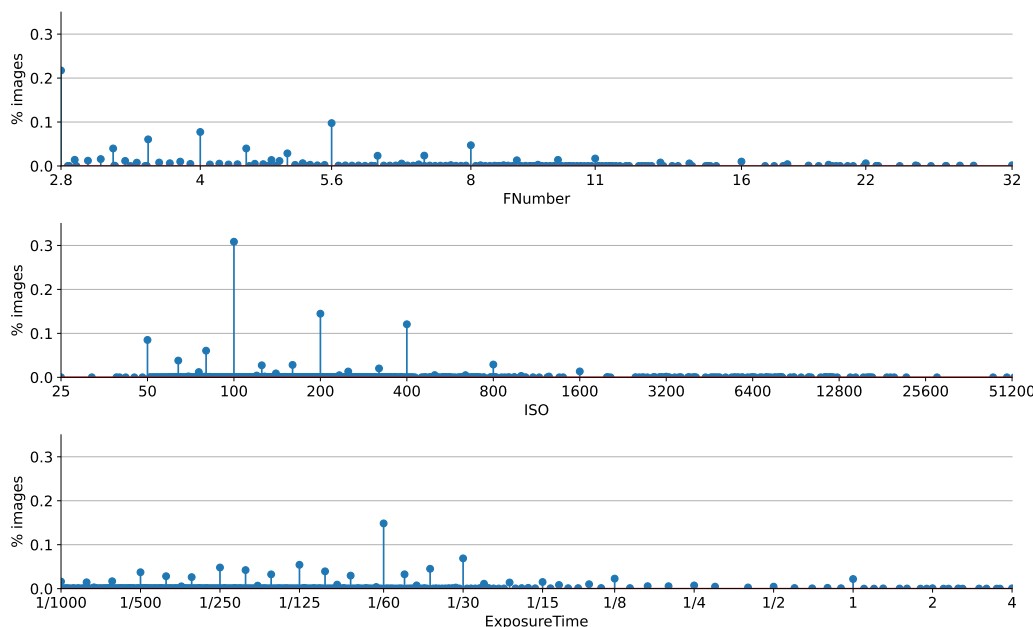

Figure A.4: Normalized distribution of camera parameter settings (F-Number, ISO, and exposure time) in the ImageNet dataset. Horizontal axis is plotted on a log-scale. The observed peaks at 1-stop intervals and very long-tailed distribution is representative of all datasets we tested.

## B    SNAP DATASET PROPERTIES

Here, we provide additional details and qualitative samples from the proposed SNAP dataset. Figure Fig. B.1 shows a general overview of the 100 unique scenes (10 for each of the 10 objects) captured in the dataset. Each of the scenes was photographed with the full range of Canon EOS Rebel T7 camera parameters (listed in Table 2) under 2 illumination conditions.

Due to exposure equivalence, many photos of the same scene look very similar, despite being taken with different camera parameters and under different level of lighting. This is illustrated in Fig. B.2. To find sets of images with the same camera settings, we estimate exposure value (EV) index from camera parameters using a standard formula from Ray et al. (2000): $EV = \log_2(\frac{\text{F}-\text{Number}^2}{\text{ExposureTime}}) - \log_2(\frac{\text{ISO}}{100})$. These EV values were used to produce plots in Fig. 1.

Exposure depends also on the illumination level, which is not reflected in this formula. Therefore, images with the same EV value but taken in different lighting conditions will look different. To bin together images that have the same exposure, regardless of illumination, we introduce EV offset (similar to exposure compensation). For each illumination condition, we find EV index that is the closest to the auto settings and assign EV offset of 0 to that bin. We compute the other EV offsets from EV relative to the 0 bin. For the 1000 lux condition we use EV of 11 as a starting point EV of 5 for the 10 lux. Now, all images with EV offset of 0 are well-exposed and those with negative or positive EV offsets are under- or over-exposed, respectively. Because camera settings in SNAP were sampled at 1-stop intervals, increase/decrease by 1 EV offset means doubling/halving of the amount of light reaching the sensor. An illustration of images with different EV offsets is provided in Fig. B.3.

## C    HUMAN EXPERIMENT

To establish a human baseline for the categorization and subitization task under different capture conditions, we tested 43 subjects (25 M, 17 F, 1 did not specify), ranging in age from 20 to 67 years old (Mean=30.2, SD=10.5), on a subset of images from the SNAP dataset. Subjects were recruited through the university mailing list and posters on the bulletin boards on campus. The study posed

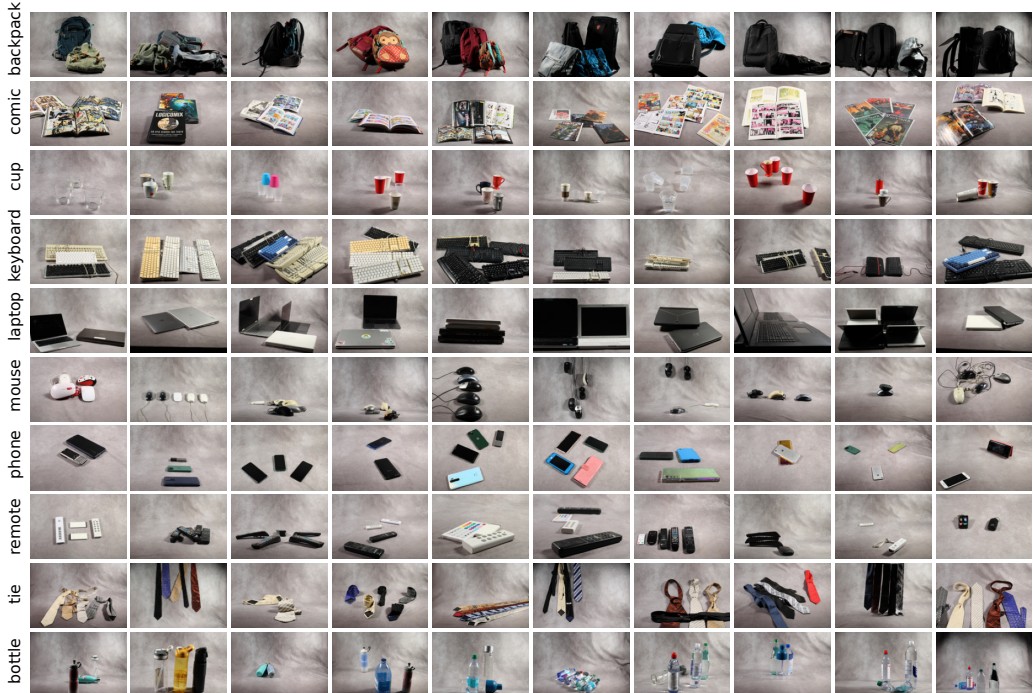

Figure B.1: Overview of the SNAP dataset. For each of the 10 object categories, we capture 10 scenes with different number of objects in different configurations. Each of these scenes is captured with the full range of camera parameters under 2 illumination conditions.

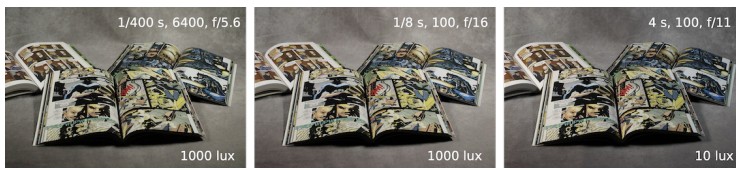

Figure B.2: An illustration of the exposure equivalence. All photos are taken with the same exposure but this is achieved with 4 different camera parameters (listed in the top-right corner) and under different illumination conditions.

minimal risk to participants and received ethics approval from the IRB (certificate # e2025-053). All subjects signed consent forms and were compensated 20CAD for their participation in the experiment, which took approximately 30 minutes.

Because our dataset consists of images with different levels of exposure, controlling display calibration and lighting conditions is crucial as it can affect what the subjects see on the screen. Thus, we conducted the in the dark room in our lab. Subjects sat 60 cm away from the calibrated 22" LCD monitor and viewed images (shown at 960×640 px resolution) on the screen. All subjects had normal or corrected-to-normal vision.

We tested the subjects on the same set of questions as the VLMs. Recalling Section 3.2.2, there were 4 questions in total: subitization (how many objects are in the image?) and categorization (what class of objects are in the image?), each with 2 versions—multiple-choice (MC) and open-ended (OE).

Subjects were randomly assigned one of two groups: group 1 answered MC subitization and OE categorization questions, group 2 answered OE subitization and MC categorization questions. This was done to prevent priming effects, i.e. subjects that had answered the MC categorization question did not answer the OE version of the same question as they would be familiar with the available options.

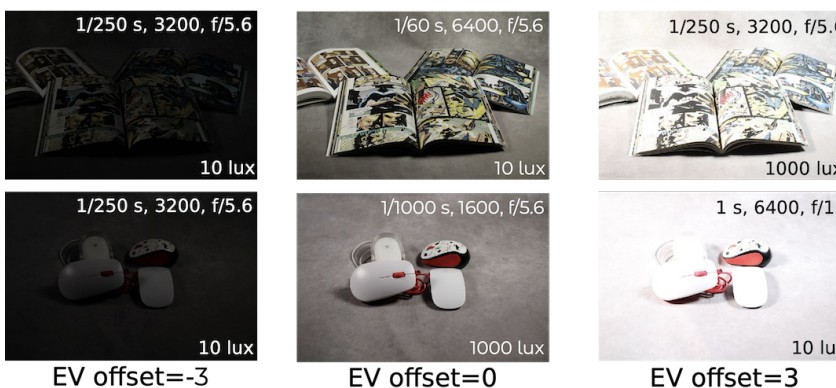

Figure B.3: Illustration of the EV offsets corresponding to different exposure levels. Bin with the "optimal" exposure (closest to the auto settings) is assigned an EV offset of 0 and the remaining bins are indexed relative to it. Negative and positive EV offsets correspond to under-exposed and over-exposed images, respectively.

Each subject performed 200 trials, 100 for each question, split into blocks of 50. Since we have 100 unique scenes in the SNAP dataset, each scene was seen twice. To minimize the chance of subjects recognizing previously seen images, we sampled pairs of images at least 4 stops apart. Furthermore, both the order of blocks and order of trials within each block were randomized. At the start of each block, the question appeared on the screen and remained the same for the all images in the block. No feedback was given to the participants during the experiment. The same questions and answer options for MC questions were used in the experiment and to test VLMs to allow direct comparisons.

Each trial proceeded as follows: 1) fixation cross was shown for 1s; 2) image appeared for 200 ms followed by a white mask for 200 ms; 3) a question prompt appeared: for open-ended question, a text box was provided where the subject could enter the answer using the keyboard; for multiple choice questions, options were shown on the screen to be selected via mouse; 4) after answering the question, subjects pressed space to continue to the next trial. The experiment was not timed and only the answers were recorded. We used PsychoPy Peirce et al. (2019) to control the experiment.

A practice session consisting of 20 trials (10 for each question) was conducted prior to the experiment to familiarize the subjects with the procedure. This session was not recorded and used images from a separate set with different object categories (tv, umbrella, spoon, orange, banana, sports ball, potted plant).

The subjects performed a total of 8600 trials with unique images from SNAP, evenly distributed w.r.t. to object categories and capture conditions.

## D  MODELS

Below we list baseline and SOTA models, including 23 image classifiers (Table D.1), 16 object detectors (Table D.2), and 13 vision-language models (Table D.3), for image classification, object detection, and visual question answering (VQA) tasks, respectively.

## E  EVALUATING VLMS

Although the expected answer for all of our questions is one word or one digit, computing accuracy for VLMs is challenging because many of them do not follow the prompt correctly. Simple matching of the keywords over-inflates accuracy because it registers a hit when the prompt is returned as part of the answer or when models consider each option in order for chain of reasoning. We manually cleaned the data to bring answers of all models to the common form. Specifically, we converted all all spelled-out numeric answers to integer format (e.g. "three" → 3) and used regular expressions to remove common answer patterns (e.g. "The objects in this image are"). These simple operations

Table D.1: Image classification models used for experiments. Model name is how the model is referred to in the text and figures. In addition, we provide the original model id and URL.

| Model name | Model id, URL |
|---|---|
| ViT-L/16-SigLIP@384 Zhai et al. (2023) | ViT-L-16-SigLIP-384 |
| SoViT-400M-SigLIP@384 Zhai et al. (2023) | siglip-so400m-patch14-384 |
| OpenCLIP ViT-L/14 Cherti et al. (2023) | vit_large_patch14_clip_224.laion2b_ft_in1k |
| DFN CLIP ViT-H/14 Fang et al. (2024) | ViT-H-14-quickgelu_dfn5b |
| DFN CLIP ViT-L/14 Fang et al. (2024) | ViT-L-14-quickgelu_dfn2b |
| ViT-MAE-L/16 He et al. (2022) | vit_large_patch16_mae |
| CLIP ConvNeXt-XXL Liu et al. (2022) | convnext_xxlarge.clip_laion2b_soup_ft_in1k |
| ConvNeXt-XL Liu et al. (2022) | convnext_xlarge.fb_in22k_ft_in1k |
| ViT-bigG-14 Radford et al. (2021) | ViT-bigG-14_laion2b_s39b_b160k |
| CLIP ViT-L/14@336px Radford et al. (2021) | clip-vit-large-patch14-336 |
| CLIP ViT-L/14 Radford et al. (2021) | clip-vit-large-patch14 |
| CLIP ViT-g/14 Radford et al. (2021) | ViT-g-14_laion2b_s12b_b42k |
| ViT-L/16-IN21K Dosovitskiy (2021) | vit_large_patch16_224.augreg_in21k_ft_in1k |
| ViT-L/16 Dosovitskiy (2021) | vit-large-patch16-224 |
| Swin-L Liu et al. (2021) | swin_large_patch4_window7_224.ms_in22k_ft_in1k |
| CSPResNet50 Wang et al. (2020) | cspresnet50 |
| CSPDarkNet53 Bochkovskiy et al. (2020) | cspdarknet53.ra_in1k |
| NASNet-L Zoph et al. (2018) | nasnetalarge.tf_in1k |
| DarkNet53 Farhadi & Redmon (2018) | darknet53.c2ns_in1k |
| ResNet101 He et al. (2016) | resnet101.tv_in1k |
| ResNet50 He et al. (2016) | resnet50.tv_in1k |
| VGG16 Simonyan & Zisserman (2015) | vgg16.tv_in1k |
| InternImage-G Wang et al. (2023) | internimage_g_22kto1k_512 |

Table D.2: Object detectors used for experiments. Model name is how the model is referred to in the text and figures. In addition, we provide the original model id and URL.

| Model name | Model id, URL |
|---|---|
| YOLOv11-XL | yolo11x |
| GroundingDINO B Liu et al. (2024b) | grounding-dino-base |
| RT-DETR R101_O365 Zhao et al. (2024) | rtdetr_r101vd_coco_o365 |
| RT-DETR-XL Zhao et al. (2024) | rtdetr-x |
| RT-DETRv2 R101 Zhao et al. (2024) | rtdetr_v2_r101vd |
| DINO Swin-L Zhang et al. (2023) | dino-swin-l |
| DINO R50 Zhang et al. (2023) | dino-r50 |
| OWL-ViT Minderer et al. (2022) | owlvit-large-patch14 |
| MR-CNN VitDet ViT-L Li et al. (2022) | mask_rcnn_vitdet_l |
| CMR-CNN VitDet Swin-L Li et al. (2022) | cascade_mask_rcnn_swin_l |
| CMR-CNN VitDet ViT-L Li et al. (2022) | cascade_mask_rcnn_vitdet_l |
| DETA Swin-L Ouyang-Zhang et al. (2022) | deta-swin-large |
| DETA R50 Ouyang-Zhang et al. (2022) | deta-resnet-50 |
| DETR R50 Carion et al. (2020) | detr-resnet-50 |
| RetinaNet R50 Liu et al. (2016b) | RetinaNet_ResNet50_FPN |
| Faster R-CNN R50 Girshick (2015) | FasterRCNN_ResNet50_FPN |
| YOLOv12-XL | yolo12x |

Table D.3: VLMs used for experiments. Model name is how the model is referred to in the text and figures. In addition, we provide the original model id and URL.

| Model name | Model id, URL |
|---|---|
| Qwen2-VL-7B Wang et al. (2024a) | Qwen2-VL-7B-Instruct |
| Qwen2-VL-2B Wang et al. (2024a) | Qwen2-VL-2B-Instruct |
| LLaVa1.6-7B Liu et al. (2024a) | llava-v1.6-vicuna-7b |
| VILA1.5-3B Lin et al. (2024) | VILA1.5-3b |
| VILA1.5-7B Lin et al. (2024) | VILA1.5-7b |
| InternVL2-2B Wang et al. (2024b) | InternVL2-2B |
| InternVL2-4B Wang et al. (2024b) | InternVL2-4B |
| InternvL2-8B Wang et al. (2024b) | InternVL2-8B |
| PaliGemma-3B@448px Beyer et al. (2024) | paligemma-3b-mix-448 |
| PaliGemma-3B@224px Beyer et al. (2024) | paligemma-3b-mix-224 |
| DeepSeek-VL-1.3B Lu et al. (2024) | DeepSeek-VL-1.3B-chat |
| DeepSeek-VL-7B Lu et al. (2024) | DeepSeek-VL-7B-chat |

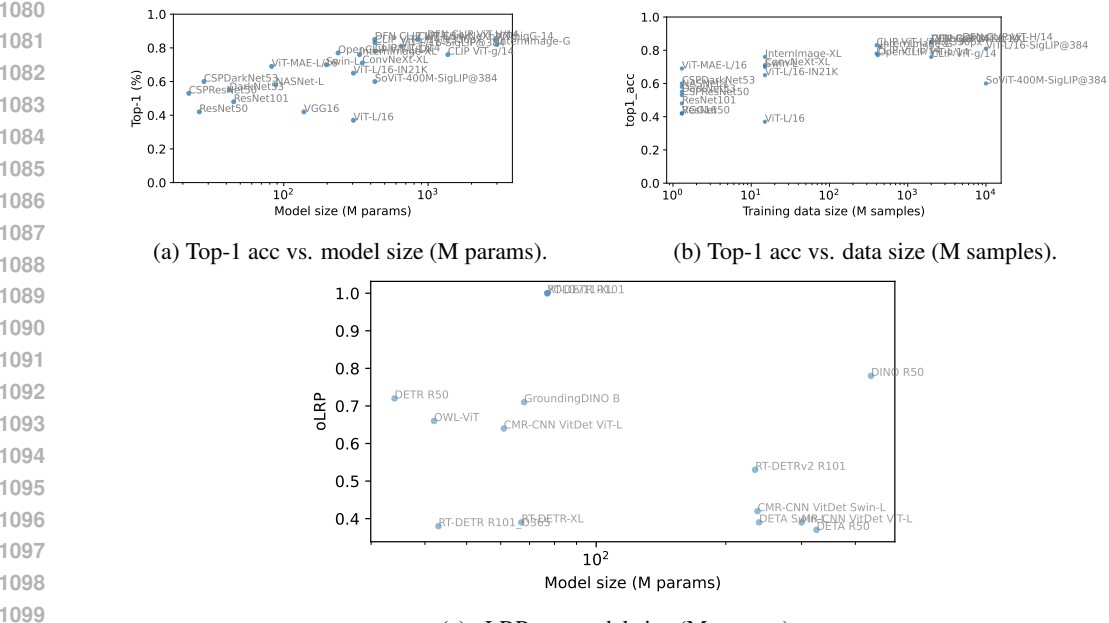

(a) Top-1 acc vs. model size (M params). (b) Top-1 acc vs. data size (M samples).

(c) oLRP vs. model size (M params).

Figure F.1: Performance of image classification and object detection models on SNAP relative to their size and training data.

cleaned up nearly 80% of the data. We examined the remaining 20% and extracted the answers by hand.

We then estimated factual errors as mismatch between the answers and the ground truth. It should be noted that this approach still likely inflates model accuracy. In many cases, the models output extra information, not relevant to the prompt. For example, when answering the open-ended categorization question, which requires just the category of the object, some models also mention the count of objects (e.g. they answer the question "Objects of what class are in the image?" with "three cups" instead of simply "cup"). For these answers, we only checked that the question in the query was answered (e.g. objects were correctly identified as cups) but not whether the additional unprompted information was correct (e.g. there could be fewer or more cups in the image).

To estimate faithfulness, we made the following checks: 1) the length of the answer (in characters) is equal or shorter than average length of the ground truth for that question (e.g. for counting questions, we expect at most 4 characters because all valid answers are numbers between 2 and 5); 2) the answer option exists (e.g. answer "E) 10" would be invalid for the multi-choice counting question because the only options are A), B), C), and D)).

Hard accuracy score for each image-question pair is assigned 1 if answer is both faithful and factual according to criteria listed above and 0 otherwise.

# F ADDITIONAL EXPERIMENT RESULTS

In this section we provide additional experimental results. Fig. F.1 contains plots that show positive correlation between performance of image classification and object detection models on SNAP and scale of the data/models. In Fig. F.2 we show that object detectors achieve better standard AP metric on SNAP than on COCO likely due to relatively simple design of SNAP, which lacks significant clutter and occlusions. Additional results for object detection task in Fig. F.3 shows a larger contribution of false negatives relative to false positives and localization errors. Because FN errors mimic patterns observed in image classification task, we speculate that the likely cause of this is the influence of image classification backbones. Lastly, Fig. F.4 shows hard accuracy of VLMs across exposures w.r.t. to human baseline and Fig. F.5 shows parameter sensitivity for each question in VQA task.

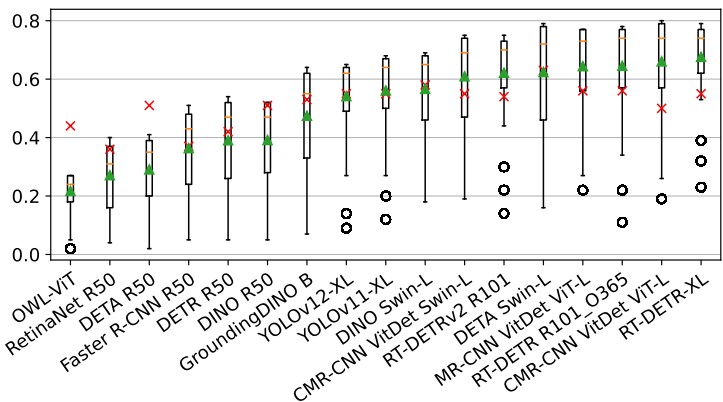

Figure F.2: Box plots show range and mean AP@[0.5:0.95] values for all object detection models evaluated on SNAP. Red crosses mark AP@[0.5:0.95] of the models on COCO. Black circles represent outliers.

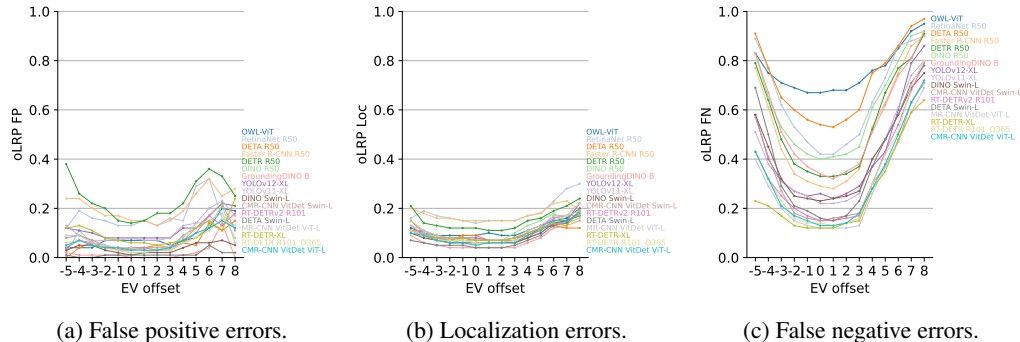

(a) False positive errors.      (b) Localization errors.      (c) False negative errors.

Figure F.3: Classification and localization errors for all exposure bins in SNAP. Each point represents a mean of the respective error for the given EV offset.

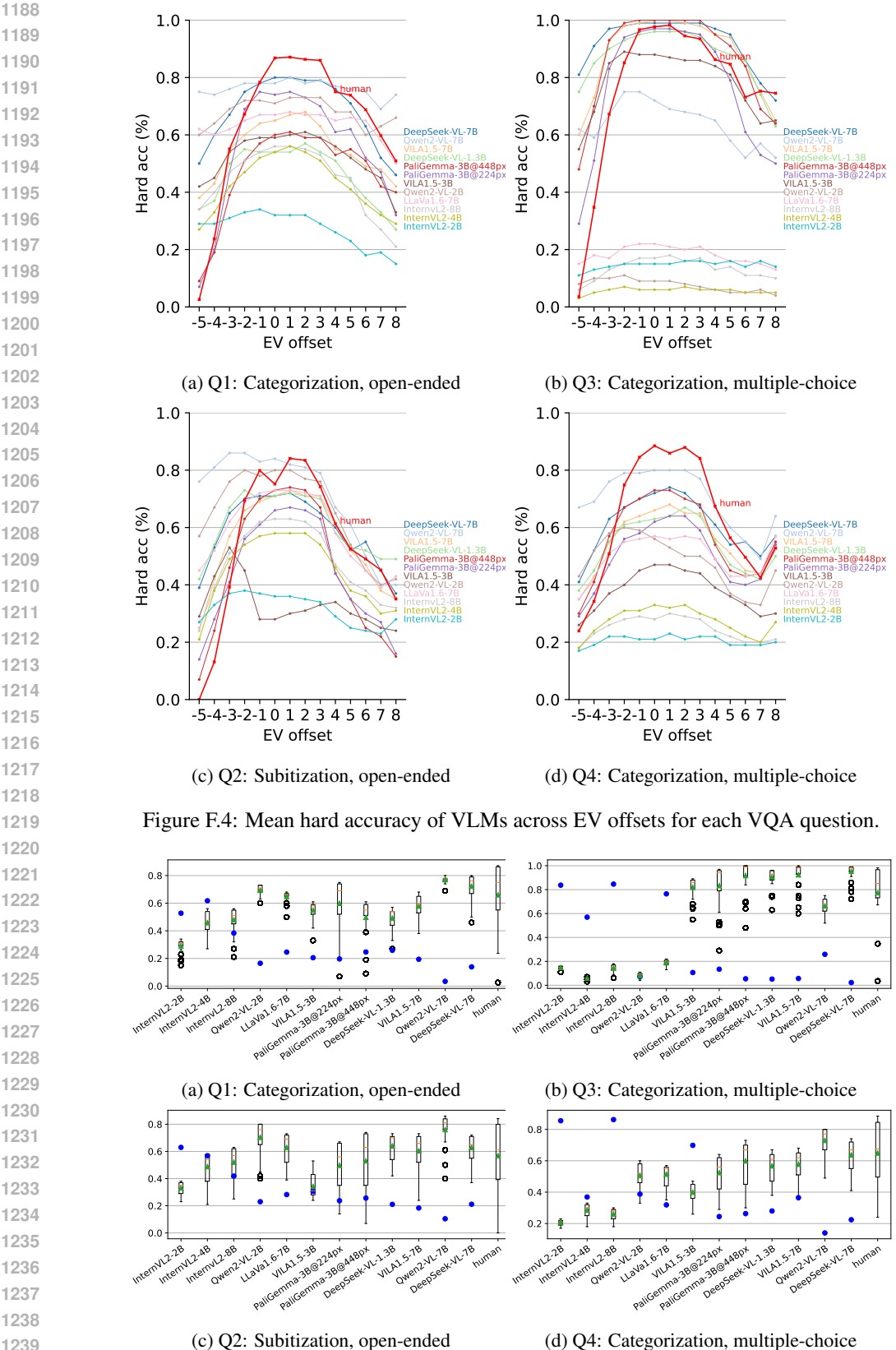

Figure F.4: Mean hard accuracy of VLMs across EV offsets for each VQA question.

Figure F.5: Box plots showing the range of hard accuracy of VLMs on each question. Parameter sensitivity (PS) is marked with blue circles. Black circles represent outliers.

