# OpenReview forum: "SNAP: Testing the Effects of Capture Conditions on Fundamental Vision Tasks"
_ICLR.cc/2026/Conference — Submitted to ICLR 2026_

### Official Review · Reviewer_g3RK · 2025-10-30

**Soundness:** 2
**Presentation:** 2
**Contribution:** 1
**Rating:** 2
**Confidence:** 5

**Summary:**

The paper introduces SNAP, a benchmark of photographs of real physical objects captured under a large grid of camera and lighting settings. SNAP evaluates three tasks—image classification, object detection, and VQA—and reports results for 52 models. The central finding is that camera parameter choices can significantly affect accuracy, highlighting practical risks when deploying vision systems outside controlled conditions.

**Strengths:**

1. **Dataset scope and design.** SNAP systematically varies sensor parameters, spanning ~700 camera‑parameter combinations, two lighting conditions, and ten object categories. It supports three representative vision tasks: image classification, object detection, and VQA.

2. **Breadth of evaluation.** The study compares 52 models across three tasks, providing a broad view of model robustness under sensor shift.

**Weaknesses:**

Despite amazing efforts for constructing the dataset and performing exhaustive experiments, my main concerns are unclear novelty compared to previous work and lack of interesting (or new) findings out of the new dataset.

1. **Unclear novelty relative to existing benchmarks.**
- The paper’s incremental contribution over ImageNet‑ES [1], ImageNet‑ES‑Diverse [2], and the SenseShift6D [3] dataset is not sufficiently articulated. These recent efforts also use physical cameras to vary lighting and sensor parameters. To strengthen the case for SNAP, the manuscript should present a clear, head‑to‑head comparison—qualitative and quantitative—throughout the introduction, related work, methodology (including Fig. 1), and experiments.

- **Qualitative difference.** A likely distinguishing aspect is that, whereas ImageNet‑ES uses displays and ImageNet‑ES‑Diverse uses printed images, SNAP captures real objects. The paper should explicitly motivate why photographing real objects offers additional scientific value beyond screens/prints and verify this experimentally. Please analyze the limitations of ImageNet‑ES and ImageNet‑ES‑Diverse and show why real‑object capture is necessary (e.g., material/geometry effects, specularities, shadows, interreflections) and how those effects manifest in performance gaps.

- **Quantitative difference.** SNAP’s denser grid (~700 vs. 64 settings in ImageNet‑ES) is a key claim. However, it is not yet clear what new insights this added density uniquely enables. The paper should demonstrate concrete findings that only emerge from a dense sampling (e.g., non‑monotonic interactions between ISO and shutter, lighting‑dependent parameter regimes, model‑specific sensitivity profiles).


2. **Limited novelty in findings.**
- The principal conclusions—(i) substantial performance degradation under sensor shift and (ii) larger models being less affected—largely echo results reported in ImageNet‑ES and ImageNet‑ES‑Diverse. The paper would benefit from emphasizing new insights (e.g., task‑specific vulnerabilities, parameter interactions, or cross‑task correlations) that go beyond prior work.

3. **Lack of solution analysis.**
- The study documents degradation but does not evaluate or discuss remedies. Do authors claim that the models should be improved for poor capture conditions where humans cannot perform well? or camera should be controlled to avoid poor capture conditions? In particular, adaptive sensing (e.g., Lens, ICLR 2025) has been shown to mitigate sensor shift in classification. An analysis on SNAP would be valuable: (i) how camera control affects performance under SNAP, (ii) whether benefits transfer from classification to object detection and VQA, and (iii) limits of such approaches in multi‑task settings.

[1] E. Baek et al., "Unexplored faces of robustness and out-of-distributions: Covariate shifts in environment and sensor domains," CVPR 2024.

[2] E. Baek et al., "Adaptive camera sensor for vision models," ICLR 2025.

[3] Y. Han et al., "SenseShift6D: Multimodal RGB-D benchmarking for robust 6D pose estimation across environment and sesor variations," arxiv, 2025.

**Questions:**

Please see the weaknesses.

---

> ### Author Response · Authors · 2025-11-17
>
> Thank you for evaluation of our work, we address concerns below.
>
> **Unclear novelty.** SenseShift6D paper was uploaded on arXiv on July 8, 2025 as we were finishing our work and was not peer-reviewed, therefore we did not make a comparison.
>
> We respectfully disagree that we needed to compare to ImageNet-ES [1] and ImageNet-ES-Diverse [2] to justify our contribution.
>
> Our contribution is not incremental to either of the datasets for the following reasons:
> - we consider a larger set of 52 models, including VLMs, while [1] or [2] consider a small subset of these;
> - we analyze model performance across 3 tasks (image classification, object detection, and VQA), whereas [1] and [2] focus on image classification;
> - we collect entirely new data, rather than recycling ImageNet or COCO, which are over-represented in the training sets of most existing models and thus may bias evaluation;
> - we provide a human baseline which contextualizes the results, whereas [2] makes broad claims about human perception without confirming them experimentally;
> - we analyze 10 major datasets comprising over 1B images that none of the other works have done;
> - we introduce a parameter sensitivity metric (PS) to study consistency of models under equivalent exposure with different camera conditions.
>
> While there is some topic overlap between our work and other past works, we use a substantially different methodology and analysis.
>
> **Qualitative difference.** ImageNet-ES has a methodological issue that the screen on which the images are displayed is itself a source of light in the scene and thus affects exposure. Having images on the flat screen also has no effect on the aperture parameter since there is no depth in the scene. We attempted to replicate ImageNet-ES in early experiments, however, it was not possible since the authors specify only the lumens of the lamps used for the illumination. A more precise measurement should have been done with a light sensor, which also would have reflected the contribution of the light emitted by the display.
>
> **Quantitative difference.** The advantage of having a dense grid is that it produces a large number of images where exposure is equivalent but lighting and camera settings change (see Fig. B2 for an example). Using a parameter sensitivity (PS) metric, we show that these largely imperceptible differences cause performance issues across models and tasks we considered. However, because ISO, shutter speed and aperture are tightly coupled and small change in either will affect exposure, it is impossible to show specific interactions between parameters. An in-depth analysis of each model performance was out of the scope of this work.
>
> **Limited novelty in findings.** We respectfully disagree with this assessment of our work. In addition to the two conclusions that appear in ImageNet-ES and ImageNet-ES-Diverse, we establish results for other tasks, provide a human baseline, a larger set of models, and connect them to our analysis of the training data. None of these conclusions could be trivially extrapolated from previous work and are thus valuable.
>
> Furthermore, we do not see a fundamental issue in confirming some of the earlier fundings. We are using a very different experimental setup and analysis, therefore this confirmation adds more support to those claims and is no less valuable.
>
> **Lack of solution analysis.** This work was focusing on the diagnostic analysis and establishing a human baseline. Solution analysis would have required additional experiments and discussion. Within the restrictive paper limit, this would have compromised the depth of the analysis for the multiple tasks and human experiment.

---

### Official Review · Reviewer_AFDT · 2025-10-31

**Soundness:** 3
**Presentation:** 3
**Contribution:** 2
**Rating:** 4
**Confidence:** 4

**Summary:**

The paper presents SNAP, a dataset designed to study how camera settings and lighting affect the performance of deep learning models on image classification, object detection, and visual question answering (VQA). The authors first analyze capture bias in 13 major datasets, finding that most images are taken under similar exposure conditions. They then build SNAP with 37k real images captured under controlled lighting and camera parameters, and evaluate 52 models along with human subjects. Results show that models are highly sensitive to even minor changes in exposure, perform worse than humans on poorly lit images, and that capture bias in training data propagates across downstream tasks.

**Strengths:**

- Thorough empirical scope: 52 models across classification, detection, and VQA.
- Novel dataset (SNAP) enabling controlled study of real capture bias.
- Human comparison adds interpretability to machine performance.
- Comprehensive dataset analysis across 1.3B images provides strong motivation.
- Clear presentation and reproducible methodology.

**Weaknesses:**

- No dense prediction tasks (e.g., segmentation, depth estimation), which would show whether pixel-level sensitivity differs from categorical tasks.
- Lack of quantitative comparison to synthetic corruptions (e.g., Gaussian noise, brightness jitter, or CSF filters) to contextualize real vs. synthetic robustness.
- The takeaways, while valid, are somewhat predictable (“models fail on extreme exposure, larger models do better”) and lack actionable insight for improving model robustness.
- The dataset, while well-controlled, is limited to 10 object categories and might not generalize beyond tabletop scenes.
- No analysis of whether findings could have been replicated using synthetic exposure shifts on ImageNet images, which would clarify the added value of real capture data.

**Questions:**

### **Questions for Authors**

1. Why were no dense tasks (e.g., segmentation/depth) included? Would these show similar exposure sensitivity?
2. Could the same conclusions be drawn using synthetic exposure perturbations applied to ImageNet or COCO (e.g., gamma correction, brightness scaling)?
3. Are there any quantitative correlations between model pretraining datasets’ exposure distributions (e.g., LAION EV histogram) and performance on SNAP?
4. Does the dataset allow evaluating data augmentation strategies that explicitly normalize exposure?


The paper is well executed and clearly written, offering valuable insights into how capture conditions affect vision models. The SNAP dataset and large-scale evaluation are solid contributions. However, the scope feels narrow—it focuses primarily on exposure, lacks dense prediction tasks, and omits comparison with synthetic corruptions, which limits its broader impact. Overall, it is a careful and meaningful study, but not particularly novel or exciting. Including a dense task and a more direct discussion of synthetic corruptions would be necessary to raise my score.

---

> ### Author Response · Authors · 2025-11-16
>
> Thank you for evaluating our work, we respond to questions below.
>
> **1. Why were no dense tasks included? Would these show similar exposure sensitivity?**
>
> We considered adding segmentation as one of the tasks and have segmentation masks in the SNAP dataset. However, we ultimately decided against it for two reasons. First, it would have been difficult to conduct a similar experiment with human participants. In short, it would be impossible to create equal conditions for fair comparisons between humans and models as segmentation is not done in one feed-forward pass by humans. Second consideration was the limited space of the paper. Additional experiments on segmentation would leave less room to properly analyze and discuss all results. However, we agree that additional tasks would be interesting for further investigation and plan to expand our work along these lines.
>
> Our preliminary results showed similar exposure sensitivity. Particularly, changing aperture introduced additional issues due to depth of field changing.
>
> **2. Could the same conclusions be drawn using synthetic exposure perturbations? **
>
> No, because as we noted in the paper, none of the synthetic datasets provided references to the real corruptions because they are difficult to establish. Qualitatively speaking, based on our observations, the “real” noise is typically much subtler than artificial noise used in the past studies. As our work shows, even these barely perceptible differences (see Figure B1 for an example) cause issues in model performance.
>
> **3. Are there any quantitative correlations between model pertaining datasets exposure distributions and performance on SNAP?**
>
> Graphs of EV offsets (Figure 4, 6, 9) can be interpreted as such quantitative correlations.  The well-exposed range in SNAP matches the conditions in most datasets. This is where the performance of most models on SNAP is comparable to results reported on other datasets. However, most images in SNAP are not well-exposed by design and likely have few if any analogs in training data. In these under- or over-exposed areas performance drops.
>
> **4. Does the dataset allow evaluating data augmentation strategies?**
>
> The dataset can be used for testing the effects of augmentation. Among the models we tested, only several performed augmentations that changed brightness and/or added noise. However, the vast majority of them did not, particularly the large-scale models. As there are no foundation models trained with augmentations for comparisons, we did not pursue this line of inquiry and instead focused on the core results of the study and comparisons to the human baseline.
>
> We also wanted to respond to two other weakensses pointed out in the review.
>
> **Takeaways are predictable.** Please see the response to reviewer g3RK regarding the contributions of the paper. In short, we use a methodology different from past studies, consider a larger set of models (including SOTA VLMs), and analyze model performance across more tasks. The conclusions we reached could not be trivially derived from the past works. The fact that we reached a similar conclusion using a different methodology is a complementary result that has value. Furthermore, we provide a human baseline for these tasks, which is likewise novel and could not have been derived from existing works. Our analysis of the common vision datasets highlights a significant bias and can be used to diversify data collection in the future. In addition, we identified a novel chronological effect due to ImageNet data being outdated and not representing appearance of the modern object appearances (see Appendix A)
>
> **Dataset generalization.** Please see the response to reviewer FYV2 regarding the object selection. In sum, even these limited scenes, that are within the OOD cause significant issues for most models but not for humans. This is one of the main points of our study. We expect that more complex scenes will cause further performance degradation and will focus on this in our follow-up work.

---

> > ### Comment · Reviewer_AFDT · 2025-11-22
> >
> > While I understand that conducting a similar experiment with human participants would have been challenging, the title of the paper is *“Testing the Effects of Capture Conditions on Fundamental Vision Tasks”*. Segmentation falls naturally under fundamental vision tasks, and I would therefore still expect it to be included, even if human performance cannot be evaluated in that setting.
> >
> > Depth estimation could also serve as such a task. A simplified formulation — for example, “Which object is closer to the camera?” — could be answered by a human, while the same question could be evaluated automatically by computing the average depth per object and ranking them. Including at least one of these tasks would make the paper substantially stronger.
> >
> > Regarding the authors’ argument that “real” noise is subtler than synthetic noise used in previous studies, I acknowledge this point. However, given that the core contribution of the paper is an analysis of the impact of capture conditions, I still believe it would be beneficial to contextualize real versus synthetic robustness by comparing the two. A minimal requirement, in my view, would be to illustrate how synthetic corruptions relate to their real-world counterparts, highlighting either the differences or, unexpectedly, the similarities.
> >
> > I thank the authors for their response, but the answer did not change my assessment. I strongly encourage including one of the two suggested experiments, because without them the impact and conclusions of the work remain somewhat limited.

---

> > > ### Author Response · Authors · 2025-11-28
> > >
> > > Thank you for the response. We want to add that in many domains, real data are used as a validation for synthetic perturbations or data from simulations. Therefore, we did not see the need for such comparisons as we were capturing real data.
> > >
> > > Regarding the task, our work expands the breadth of the previous works that focus almost exclusively on image classification. It would not be feasible to add more tasks without compromising the quality of analysis and discussion.

---

### Official Review · Reviewer_FYV2 · 2025-11-01

**Soundness:** 2
**Presentation:** 2
**Contribution:** 2
**Rating:** 2
**Confidence:** 4

**Summary:**

In this paper, the authors study the effects of capture conditions (i.e., camera parameters and lighting) on three common vision tasks (i.e., image classification, object detection, and VQA). To achieve this, the authors first construct a new dataset under controlled lighting conditions and with densely sampled camera settings. The authors then conduct an empirical study by evaluating a large number of vision models on the created dataset and reveal the effects of capture conditions on each selected vision task.

**Strengths:**

-	The idea of studying the effects of image capture conditions on the generalizability of models is interesting.
-	The authors reveal some interesting observations based on their created dataset. The evaluation is comprehensive, covering a large number of models.

**Weaknesses:**

-	My major concern is that the created dataset is less practical in real-world scenarios. 1) The dataset only contains 10 categories, most of which are office supplies and small-scale objects. However, there are many other categories in ImageNet and COCO. Using this small set of categories tends to make the evaluation somewhat biased. 2) The scenes are not cluttered, objects appear large against a plain background, and there is little occlusion. Such a toy case makes me concerned about the actual usefulness of the dataset.
-	I am curious what if the models are fine-tuned on the dataset. Will the models still suffer from under-exposed and over-exposed conditions? To do this, the authors can split the dataset into a training set and a test set. Then the authors can finetune the models on the training set and evaluate them on the test set.

**Questions:**

I am concerned about the questions mentioned above. Given the current status of the paper, I am leaning towards rejection and hope the authors could address my concerns during the rebuttal.

---

> ### Author Response · Authors · 2025-11-16
>
> Thank you for your evaluation of our work, we address the questions below.
>
> **Practicality of the dataset.** The dataset we created is meant as a diagnostic test that measures performance of the models across systematically varied capture conditions. As we noted in the paper, there were several reasons for choosing these 10 object categories:
> - each object is captured across hundreds of camera settings for 2 light conditions, which is a time-consuming and labor intensive process.
> - we chose categories that overlapped across COCO and ImageNet to enable testing models trained on either or both datasets without collecting more data
> - there were additional limitations on the object categories: 1) only inanimate objects were chosen to ensure that the scene remains exactly the same during the long capture times and there are no effects such as motion blur; 2) the objects had to fit within our capture setup, which limited their physical size, 3) outside scenes were excluded due to the inability to guarantee both the static scene and consistent lighting.
>
> The scenes themselves were made intentionally simple with little clutter and little occlusion. The scenes are also similar (but not identical) to some of the samples from the ImageNet. This is to further isolate the effects of the capture conditions themselves. If we had significant clutter and/or occlusions, then dips in performance could be attributed to these factors. Yet, even under these simplified conditions we found that even the SOTA models do not reach human performance under optimal capture conditions. We intend to expand this work to more complex scenes, where we anticipate further performance degradation across the board.
>
> **Fine-tuning.** The dataset has 100 unique scenes (10 per object category). We expect that fine-tuning will lead to some improvement but overfitting will be difficult to control, therefore such experiment will have little value.
>
> To summarize, our dataset is meant as an experimental validation of the model performance that to the extent possible isolated capture conditions from other influences on model performance. Another contribution is a human baseline for the same tasks. Due to the labour-intensive data collection, the scale of the data is not sufficient for training/fine-tuning models, however, is useful as a diagnostic, similar in spirit to ObjectNet (Barbu et al, NeurIPS, 2019)

---

### Meta-Review · Area_Chair_AxS1 · 2026-01-04

**Summary:**

This paper introduces SNAP, a dataset for studying how capture conditions affect image classification, object detection, and VQA. Reviewers praise the clear writing, substantial data collection, thorough evaluation, and inclusion of a human baseline. However, the contribution is seen as incremental: the dataset covers only 10 object categories in simplified scenes, evaluation is limited to three tasks, and novelty over existing benchmarks is unclear. The rebuttal clarifies design choices but does not address these core concerns. Overall, the recommendation is to reject.

**Reviewer Concerns:**

The rebuttal clarifies the dataset’s diagnostic intent and design choices, partially addressing reviewer concerns. However, major issues remain, including limited differentiation from related benchmarks, narrow task coverage without dense prediction tasks, missing comparison to synthetic perturbations, and largely confirmatory findings with limited new insight.

**Reviewer Scores:**

The paper received two rejects and one borderline reject and the rebuttal does not change the core concerns so the scores would likely remain unchanged.

---

### Decision · Program_Chairs · 2026-01-26

Reject